# Automatic Extraction Method of Aquaculture Sea Based on Improved SegNet Model

Weiyi Xie [1], Yuan Ding [1], Xiaoping Rui [1,*], Yarong Zou [2,3] and Yating Zhan [4,*]

1   School of Earth Sciences and Engineering, Hohai University, Nanjing 211100, China;
    xiewy@hhu.edu.cn (W.X.); dingyuanhhu@hhu.edu.cn (Y.D.)
2   National Satellite Ocean Application Service, Ministry of Natural Resources, Beijing 100081, China;
    zyr@mail.nsoas.org.cn
3   Key Laboratory of Space Ocean Remote Sensing and Application, Ministry of Natural Resources,
    Beijing 100081, China
4   Geological Survey of Jiangsu Province, Nanjing 210018, China
*   Correspondence: ruixp@hhu.edu.cn (X.R.); zytayx@sina.com (Y.Z.)

**Abstract:** Timely, accurate, and efficient extraction of aquaculture sea is important for the scientific and rational utilization of marine resources and protection of the marine environment. To improve the classification accuracy of remote sensing of aquaculture seas, this study proposes an automatic extraction method for aquaculture seas based on the improved SegNet model. This method adds a pyramid convolution module and a convolutional block attention module based on the SegNet network model, which can effectively increase the utilization ability of features and capture more global image information. Taking the Gaofen-1D image as an example, the effectiveness of the improved method was proven through ablation experiments on the two modules. The prediction results of the proposed method were compared with those of the U-Net, SegNet, and DenseNet models, as well as with those of the traditional support vector machine and random forest methods. The results showed that the improved model has a stronger generalization ability and higher extraction accuracy. The overall accuracy, mean intersection over union, and F1 score of the three test areas were 94.86%, 87.23%, and 96.59%, respectively. The accuracy of the method is significantly higher than those of the other methods, which proves the effectiveness of the method for the extraction of aquaculture seas and provides new technical support for automatic extraction of such areas.

**Keywords:** aquaculture sea; deep learning; pyramid convolution; CBAM





## 1. Introduction

The term "aquaculture sea" refers to an area in marine waters or coastal zones, such as open ocean, bays, rivers, etc. [1–4], where aquaculture is performed. As an important component of aquaculture, aquaculture sea is of great significance because it enriches the diversity of human consumption and promotes development of the marine economy. In addition, it can promote the development of marine science and technology and marine ecological environment protection and sustainable development [5–8]. Due to the malleability of the aquaculture industry, the area and type of aquaculture sea change every year. Therefore, timely, accurate, and efficient extraction of aquaculture sea is very important for the scientific and rational use of marine resources and the protection of the marine environment.

The continuous development and updates in remote sensing technology provide a constant impetus for the extraction of aquaculture sea areas. The increasing availability of multi-platform, multi-type, and multi-resolution remote sensing data offers more choices for extracting aquaculture sea areas. Long-distance satellite remote sensing saves time, man-power, and costs compared to on-site surveys, offering advantages such as efficiency, wide coverage, and sustainability [9–13]. However, it also has limitations. Remote sensing-based

extraction techniques are somewhat dependent on factors like climate, temperature, and season, and are constrained by factors like satellite imaging modes, acquisition timing, and data processing accuracy. In addition, although various remote sensing-based extraction techniques are available, they all have room for improvement. The challenges of "same spectral characteristics, different matter" and "same substance, different spectrum" need urgent solutions. Additionally, aquaculture sea areas vary significantly, ranging from large, concentrated regions to small, dispersed areas, with substantial regional differences and uneven distribution. For such complex areas, the precision of remote sensing extraction needs enhancement, and a more holistic approach is required for the extraction of large-scale aquaculture sea areas.

Therefore, based on remote sensing imagery of the Gaofen-1D satellite and to improve the automatic extraction accuracy of aquaculture sea and provide technical support for scientific monitoring and management of aquaculture sea, this study constructed an improved SegNet model by introducing the pyramid convolution module and attention mechanism module. In this study, we optimized the SegNet model. The introduction of the pyramid convolution module has enhanced the model's ability to extract features across different spatial scales, significantly improving the precision of aquaculture sea area delineation. Simultaneously, the incorporation of the attention mechanism module has played a crucial role, directing the model's attention to essential elements within the images, resulting in a substantial improvement in the detection performance of aquaculture sea areas. These meticulous enhancements collectively equip our model with the flexibility required to effectively adapt to the complexities and diversities inherent in various aquaculture regions.

## 2. Related Work

The commonly used remote sensing image information extraction methods include manual visual interpretation classification, object-oriented classification, pixel-based classification, and deep learning-based classification [14]. The manual visual interpretation method, which relies mainly on human resources and human experience to identify features based on regional data and the characteristics and spatial relationships of features identified in remote sensing images or in the field, is the most commonly used method. He et al. [15] extracted information on cage culture in the inland waters of the Hunan Province based on Gaofen-2 remote sensing images using the human–computer interactive visual interpretation method. Ji et al. [16] designed a new algorithm for purse seine extraction that combined spectral and textural features from manually interpreted images and accurately extracted the temporal and spatial distribution information of the purse seine area of Yangcheng Lake. However, most of the aquaculture sea areas in China have a wide scope and a large area, and traditional manual field interpretation is slow, costly, and subjective.

The object-oriented classification method mainly sets a segmentation parameter according to the image information to segment the remote sensing image and thus form multiple objects, which are used as the minimum processing unit of classification. Xu et al. [17] segmented SPOT5 satellite remote sensing imagery at different scales and combined the spectral, shape, and semantic features to realize identification and extraction of aquaculture ponds. Wang et al. [18] combined association rule classification and the object-oriented method to accurately identify the four aquaculture modes of pond, cage, beach, and floating raft in complex coastal areas. Based on the Google Earth Engine (GEE) platform and Sentinel-1 and Sentinel-2 time series remote sensing data of the whole year, Wen et al. [19] combined the threshold method and the object-oriented classification method to extract the aquaculture sea in the coastal zone of Guangxi Beibu Gulf in 2019. The object-oriented classification method has some limitations. Its accuracy depends heavily on the segmentation scale and parameters, but it is often difficult to determine the optimal value and it needs to be adjusted repeatedly.

The pixel-based classification method takes the pixel as the smallest processing unit in its recognition and classification of ground objects and uses spectral, textural, spatial correlation, and other information to distinguish and classify each pixel. The common methods include the index method and the machine learning supervised classification method. Duan et al. [20–22] used the traditional machine learning decision tree method to classify Landsat satellite images, obtained the spatiotemporal changes of coastal aquaculture waters in China in the past 30 years, and analyzed the development trend and driving factors, such as geographical conditions, socioeconomic factors, and development policies. Kang et al. [23] extracted the aquaculture targets in Liaoning Province from 2000 to 2018 by performing band ratio normalization calculations on Landsat data. Wang et al. [24] extracted raft aquaculture in Luoyuan Bay in Fuzhou based on the significance normalized vegetation index. Hou et al. [25] used the combination of normalized vegetation index and support vector machine classification to monitor the dynamic characteristics of the algae culture area in Weihai City, Shandong Province. Xing et al. [26] extracted the algae culture area by using the differential vegetation index and revealed that it has no inevitable relationship with the formation of green tide in the Yellow Sea by retrieving the spatiotemporal development process through remote sensing, which has great significance to guiding the spatial planning of mariculture and the prevention and control of green tides in the Yellow Sea. Wang et al. [27] extracted the aquaculture sea area based on the GEE framework and random forest model and determined the dynamic pattern and driving factors of mariculture. However, the traditional index method and machine learning supervised classification method cannot escape the phenomena of "same spectral characteristics, different matter" and "same substance, different spectrum" and cannot automatically extract the original feature information. The accuracy of information acquisition needs to be improved. It remains difficult to provide detailed references and strong support for marine resource management.

With the rapid development of science and technology in recent years, many emerging technologies have been developed. As a branch of machine learning, deep learning performs well in target recognition, image segmentation, artificial intelligence, and more. Chen et al. [28] extracted and compared multitemporal high-resolution remote sensing images of aquaculture sea based on DeepLab-v3+ and U-Net neural network structures. Zou et al. [29] constructed a U2-Net network model to extract aquaculture sea from remote sensing images in the coastal zone of Zhoushan Archipelago, Zhejiang Province, China. Lu et al. [30] improved the U-Net network by using the cavity space convolution pool pyramid and the upsampling structure, thus reducing the edge "adhesion" phenomenon of aquaculture sea identification in the medium-resolution remote sensing image. However, in general, deep learning has fewer applications in the extraction of aquaculture sea information [31,32], focuses more on the extraction of specific information such as aquaculture rafts and cages [33–36], and produces fewer research results of large-scale automatic extraction of aquaculture sea. At the same time, the existing methods remain limited in areas with varying scales and an uneven spatial distribution, and there are problems of false extraction and missing extraction in large and dense areas and small and scattered areas of aquaculture sea.

## 3. Materials

### 3.1. Study Area

The study area was Lianyungang City, Jiangsu Province, China. The geographical coordinates range from $34°18'21''$ N to $35°1'3''$ N and from $118°41'38''$ E to $119°33'44''$ E. Known for its developed marine economy and abundant marine resources, Lianyungang is an important marine fishery area in Jiangsu Province. It has a marine area of 6677 km$^2$ and 110,000 ha of shallow tidal flats. Seventeen major rivers flow into the sea along the coast, and the water quality in the sea area is fertile. Lianyungang includes Haizhou Bay Fishery, one of the eight major fishing grounds in the country, and Huaibei Salt Field, one of the four major sea salt production areas in China. The main aquaculture crop is

seaweed, and the largest seaweed aquaculture and processing base in China is located here. In 2021, the China Fisheries Association approved the title of "China's Seaweed Capital" for Lianyungang. The aquaculture sea areas are mainly distributed in counties and cities along the eastern coast, such as Guanyun, Donghai, and Haizhou. Guanyun County has the largest-scale aquaculture sea areas. Aquaculture seas in the study area are rich in resources and complex in distribution, which is an important factor for us to choose it. The geographical location and remote sensing image of the study area are shown in Figure 1.

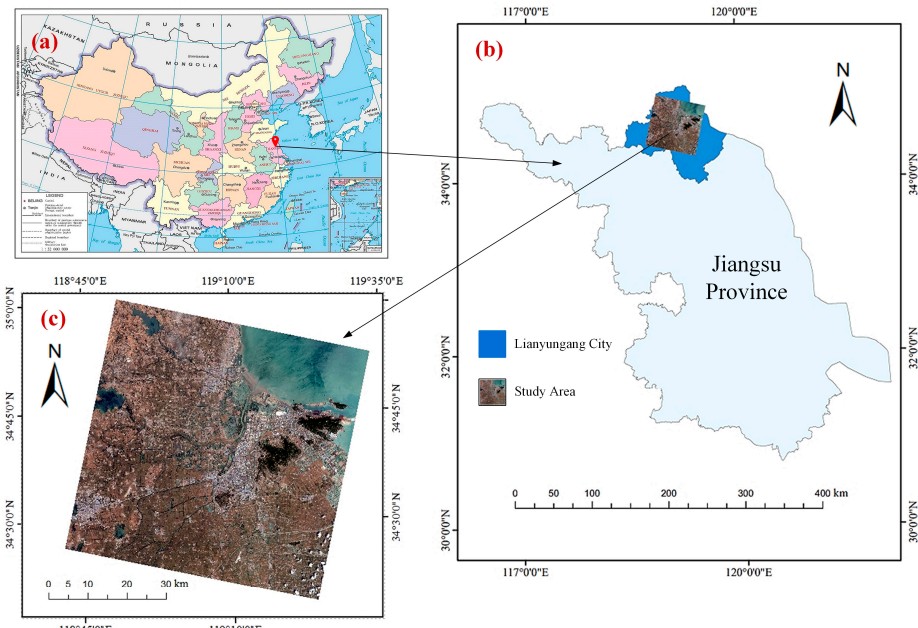

**Figure 1.** Geographic location and remote sensing image of the study area. (**a**) Map of China (Map review no. of China: GS (2019) 1681); (**b**) Location map of the study area; (**c**) Remote sensing image of the study area.

### 3.2. Data Source

The experimental data of this study were based on the L1A class remote sensing image data of the Gaofen-1D satellite of Lianyungang City and were acquired on 24 May 2022, at 11:39 LT. The Gaofen 1D satellite, launched in 2018 with B and C satellites in a "one rocket, three star" mode, has a panchromatic image spatial resolution of 2 m, multispectral image spatial resolution of better than 8 m with four bands of red, green, blue, and near-infrared, and a single satellite imaging width of greater than 60 km, which greatly improve its ability to monitor natural resources in all weather, with full coverage, and in real-time.

### 3.3. Data Preprocessing

ENVI 5.3 software was initially utilized to perform atmospheric correction and orthorectification on the high-resolution Gaofen-1 multispectral imagery and panchromatic imagery of the study area. Subsequently, the corrected images were fused to enhance the spatial resolution while preserving the spectral characteristics. The fused high-resolution multispectral image was then used to extract band 4 (near-infrared band), band 2 (green band), and band 1 (blue band), corresponding to the red, green, and blue channels, respectively. Then, ArcGIS 10.8 software was employed in conjunction with visual interpretation and manual annotation to label sample areas within the study region as either aquaculture sea or other land features. These two categories were assigned different field values, and the raster images were converted to grayscale, with a value of 1 for aquaculture sea and 0 for other land features. The aquaculture sea areas were marked as white, with an RGB value of (255, 255, 255), while the other land features were marked as black, with an RGB

value of (0, 0, 0). Finally, the vector labels were converted to raster data, completing the generation of the aquaculture sea label dataset.

After the labeling process, Python code was used to perform sliding window cropping on the entire image and its corresponding labels, resulting in uniform crops of 256 × 256 pixels. Data augmentation was then applied to the cropped samples to prevent overfitting during network training. Augmentation techniques such as horizontal and vertical rotations, diagonal flipping, and the addition of salt-and-pepper or Gaussian noise were employed to enhance the dataset. Consequently, 10,000 sample pairs, each consisting of a 256 × 256 pixels image and its corresponding label, were generated. Among these pairs, 7500 were allocated to the training dataset, and 2500 were assigned to the validation dataset. Each group of sample pairs in the aquaculture sea dataset was composed of images and corresponding tag images, as shown in Figure 2.

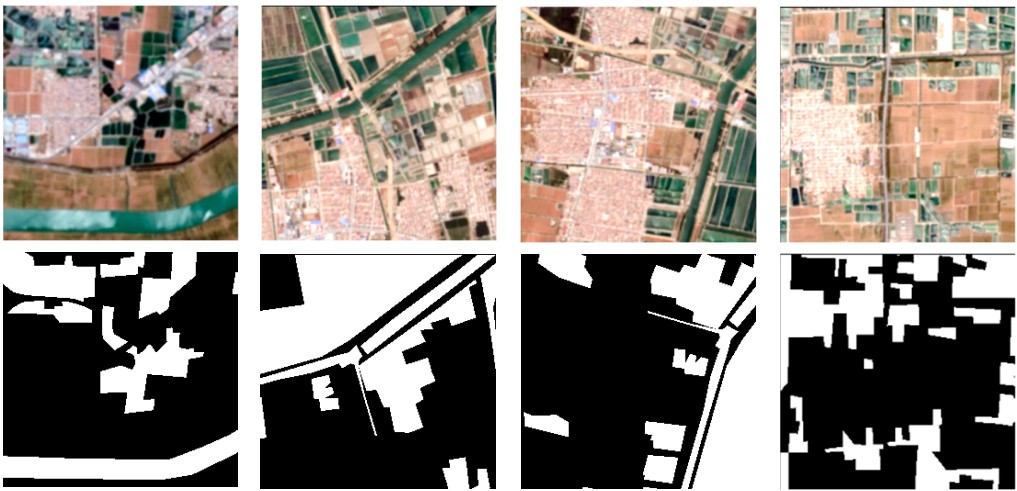

**Figure 2.** Gaofen-1D sample set of images and their corresponding labels. The top row shows the Gaofen-1D sample set of fused images, and the bottom row represents the true labels corresponding to the sample set of fused images.

Furthermore, three test regions were selected from areas outside the training images. These regions were cropped into 1000 × 1000 pixels remote sensing images, and their corresponding aquaculture sea labels were created. This formed three sets of test region pairs for comparison.

## 4. Methodology

### 4.1. Framework of the Proposed Improved SegNet Model

The goal of our research was to improve the basic SegNet network. The SegNet network, which was published in IEEE [37] by Vijay and Alex in 2017, is a basic convolutional neural network with a classic encoder–decoder structure. The network structure is clear and can be used for real-time applications quickly and with small storage space.

We developed an improved SegNet model based on the classic network structure by adding a pyramidal convolution (PyConv) module and a convolutional block attention module (CBAM), as shown in Figure 3, to strengthen the use of feature information.

First, we replaced the first standard convolution of each layer of SegNet with a pyramid convolution and designed the pyramid convolution module portion, as shown in the red box in Figure 3. In the convolution process, taking the depth of the input characteristic map as an example, the design input characteristic map first passes through 1×. The standard convolution of 1 is adapted to a channel depth of 64, and then it is convoluted by four characteristic pyramids with different convolution kernel sizes. Based on traditional experience and parameter debugging, the convolution kernel sizes of the four layers in the pyramid were set to 9 × 9, 7 × 7, 5 × 5, and 3 × 3. The groups were set to 16, 8, 4, and 1 to perform grouping convolution. Each layer generated 16 feature maps, and 64 feature maps

were generated as the output of the four layers. Subsequently, a 1 × standard convolution of 1 was adapted back to the channel depth of 256. Simultaneously, the batch normalization (BN) layer and rectified linear unit (ReLU) activation function were added after each convolution. Finally, through a quick connection, the output and input feature maps were added as the final outputs.

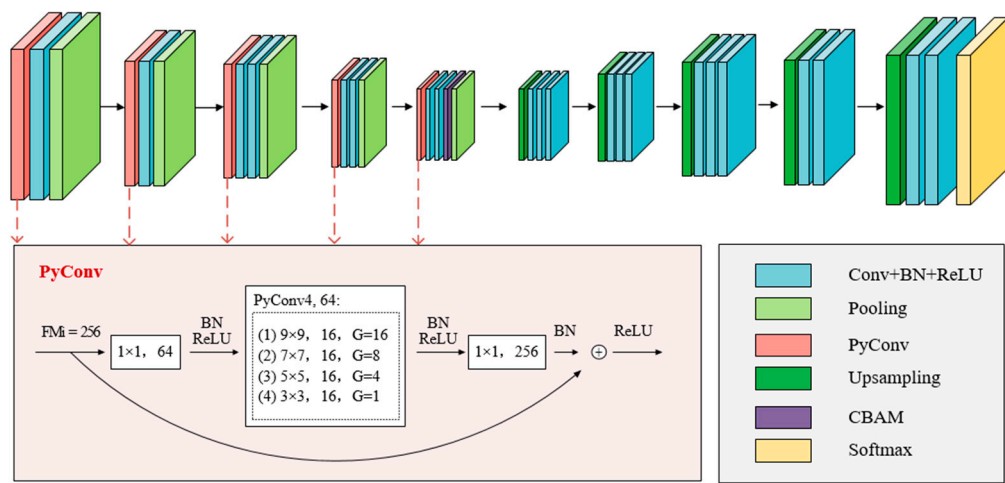

**Figure 3.** Framework of the proposed network model, the improved SegNet model. The PyConv module added in this study is represented by the red box in the figure. The gray box explains the layers in detail. The purple layer is the CBAM module added in this study.

Second, we improved the encoder of SegNet by adding a CBAM module before the pooling operation of the layer 5 network so that the input feature map successively passed through the channel attention and spatial attention modules, thus enhancing the sensitivity of the information attention to the features extracted via the encoder. According to the empirical principle and debugging, the layer 5 network belongs to the encoder part, and adding the CBAM module here reduces the phenomenon of overfitting while not causing a substantial increase of semantic information. The system can focus on the more critical information at present and reduce attention to other information to obtain more information related to the identified target and improve the utilization efficiency of the feature information.

### 4.2. Pyramid Convolution

To address the problem of the standard convolution with a single scale not being able to capture the information of complex objects well, Duta et al. [38] proposed the pyramid convolution module. Its core idea is to use different levels of convolution cores, that is, cores with different sizes and depths, to process the input image, which can better capture the details of different levels and scales, to compensate the lack of multiscale processing ability of the standard convolution. As shown in Figure 4, the pyramid convolution contains N-level pyramids with different convolution cores. From the bottom (level 1) to the top (level n) of the pyramid, the size of the convolution core increases, while the depth of the convolution core decreases.

Because of this structure, the biggest advantage of pyramid convolution is that it can realize multi-scale processing through diversified combinations. Different convolution kernels can contain both large receptive fields and small receptive fields and can focus on details while also focusing on larger objects. Moreover, the pyramid convolution does not add additional network parameters. Compared with the standard convolution, the model parameters and requirements of similar quantity levels are maintained by default in the computing resources.

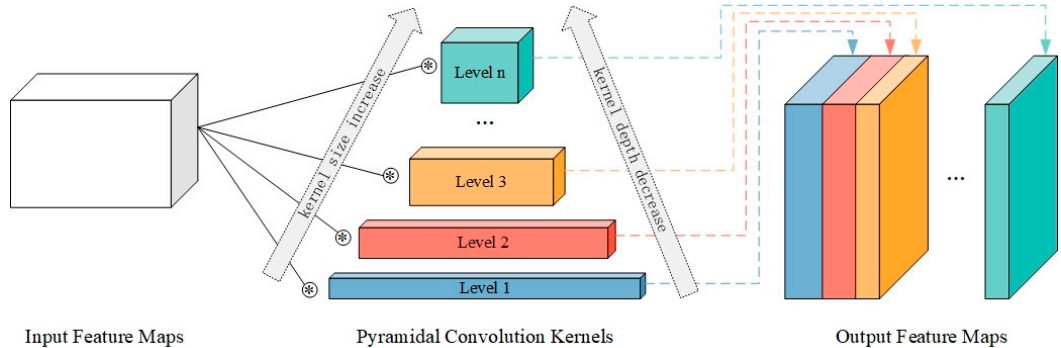

**Figure 4.** Structure of the PyConv module. The different levels form a pyramid shape, the size of the convolution core gradually increases from the bottom to the top of the tower, and the depth of the convolution core gradually decreases.

### 4.3. Convolutional Block Attention Module

The attention mechanism is a module that is often added to a convolutional neural network. Its core objective is to make the network pay attention to more important information. It generally includes two types of mechanisms: a spatial attention mechanism and a channel attention mechanism. In 2018, Woo et al. [39] proposed a lightweight convolutional block attention module (CBAM), which realized the combination of spatial attention and channel attention. Its structure is shown in Figure 5. The channel attention module and spatial attention module are processed successively for the characteristic diagram of the input network.

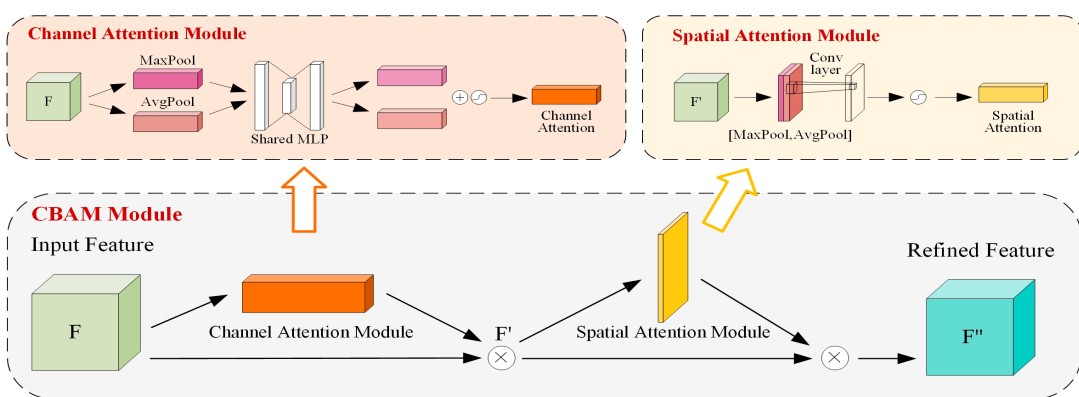

**Figure 5.** Structure of the CBAM module. It is composed in order of channel attention and spatial attention modules.

The channel attention module extracts the characteristics of each channel through global maximum pooling and global average pooling and then learns the channel attention weight by using the shared full connection layer and sigmoid activation function. As a supplement, the spatial attention module calculates the spatial attention via maximum pooling and average pooling of the features of each spatial location and stacks them. Then, it uses the standard convolution with a number of channels of 1 to connect and obtains the weight of spatial attention through the sigmoid activation function. Finally, the outputs of the two modules are multiplied and weighted to obtain the final output.

### 4.4. Experimental Platform and Parameter Settings

The experimental platform utilized a 64-bit Windows 10 Professional operating system. The system was equipped with an Intel Core i7-12700 12th generation processor, 48 GB of DDR4 3200 MHz memory, and an NVIDIA GeForce RTX 3060 graphics card. The experimental environment was configured using Anaconda3 software, creating a virtual en-

vironment with Python 3.6 for conducting the experiments. The deep learning framework was TensorFlow 2.4, with its integrated Keras 2.4 interface. To leverage the computational power of the GPU for complex calculations, the corresponding CUDA 11.1 platform was configured, along with the cuDNN 8.0 neural network acceleration library. These configurations ensured improvement of the GPU's ability to solve complex computing problems. Additionally, PyCharm 2022 software was installed as an integrated development environment (IDE) for programming, debugging, and development, guaranteeing a smooth experimental workflow.

Obtained through multiple rounds of parameter tuning experiments and optimization, the final hyperparameter settings for training the improved SegNet model are presented in Table 1.

**Table 1.** Specific parameters.

| Epoch | Batch Size | Learning Rate | Optimizer |
|:---:|:---:|:---:|:---:|
| 200 | 16 | 0.00001 | Adam |

*4.5. Precision Evaluation Metrics*

The accuracy evaluation in this study referred to comparing the identified aquaculture sea areas in the three test regions with their actual labels to assess the effectiveness and accuracy of the proposed method. The evaluation metrics relied on a confusion matrix for quantitative analysis of the aquaculture sea classification. Because the predicted images were classified into two categories, aquaculture sea and other land features, the confusion matrix took the form of a $2 \times 2$ matrix. There were four possible combinations of predicted results and real results in the matrix: TP, FP, FN, and TN, where T and F represent correct and incorrect and P and N represent 1 and 0, respectively. Among these combinations, TP represents correctly identified sea pixels for breeding, FP represents wrongly recognized sea pixels for farming, TN represents other correctly recognized pixels of other features, and FN represents other wrongly recognized pixels of other features. In this study, the number of classes, denoted as n, was 2.

Five evaluation metrics were selected in this study: precision, recall, overall accuracy (OA), F1$_{\text{score}}$, and mean intersection over union (mIoU). The formulae for these metrics are shown in Equations (1)–(5). Precision represents the probability of correctly identifying samples predicted as aquaculture sea out of all samples predicted as such. Recall denotes the probability of correctly identifying aquaculture sea samples out of all true aquaculture sea samples. Overall accuracy (OA) is the probability that the predicted result matches the ground truth label for each random sample. The F1$_{\text{score}}$ is a balanced measure that simultaneously maximizes precision and recall for a classification model. Intersection over union represents the ratio of the intersection to the union of predicted class samples and actual class samples, and mean intersection over union is the average result obtained by averaging across all classes.

$$\text{precision} = \frac{TP}{TP + FP} \tag{1}$$

$$\text{recall} = \frac{TP}{TP + FN} \tag{2}$$

$$\text{OA} = \frac{TP + TN}{TP + TN + FP + FN} \tag{3}$$

$$\text{F1}_{\text{score}} = 2 \times \frac{TP}{TN + 2TP + FP} \tag{4}$$

$$\text{mIoU} = \frac{1}{n} \times \sum_{i}^{n} \frac{TP}{TP + TN + FP} \tag{5}$$

## 5. Results

To validate the performance and demonstrate the superiority of the proposed improved SegNet model, the trained model was saved, and the three selected test areas were subjected to prediction and recognition. The predicted images of the test areas were visually analyzed and quantitatively evaluated by comparing them with the truth-label images. In this study, the proposed improved method was compared with the results obtained from the classical SegNet, U-Net, and DenseNet networks, as well as traditional machine learning methods such as support vector machines and random forests. The comparative visualizations of the original images, ground truth labels, and recognition results generated with the various methods for the three test areas are shown in Figure 6. In the images, white represents aquaculture sea areas, and black represents other land features.

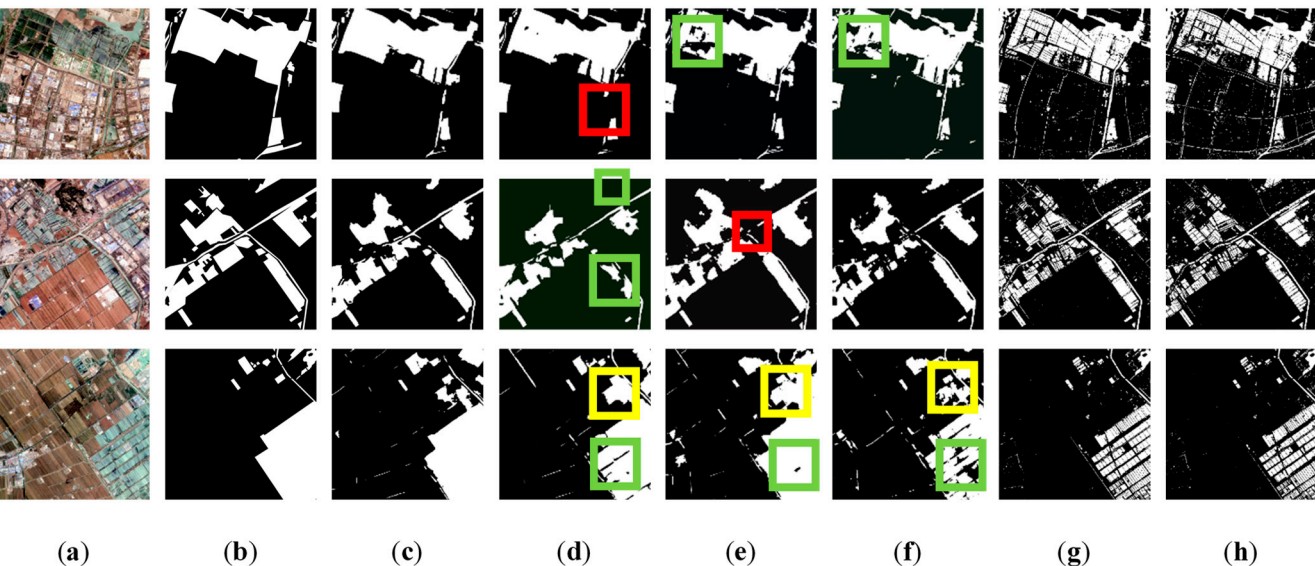

**Figure 6.** Comparison of identification results in the test area: test area 1 (row 1), test area 2 (row 2), and test area 3 (row 3). (**a**) Test images; (**b**) True labels; (**c**) Ours; (**d**) SegNet; (**e**) U-Net; (**f**) DenseNet; (**g**) Support vector machine classifier; (**h**) Random forest.

Visual comparison shows that the proposed model in this study exhibits the best visual performance, significantly reducing both misclassification and omission compared to the other models. Among the other models, the green boxes represent cases of aquaculture sea omission, which indicate a more severe omission issue within large coastal areas. The yellow boxes represent cases of misclassification of other land features as aquaculture sea, which the proposed model effectively prevents. The red boxes indicate instances of omission in identifying aquaculture sea drainage channels, for which the proposed model shows better capabilities in recognizing finer river channels than the other models.

Furthermore, the results obtained using traditional machine learning show evident fragmentation with respect to aquaculture sea identification. The recognition results for aquaculture sea are obviously more scattered and fragmented, showing poor extraction performance and lower regional integrity and coherence. Through the analysis, it was concluded that traditional machine learning methods rely solely on single-dimensional information, such as spectral data, and primarily focus on individual pixel-wise feature judgments. They lack the ability to interpret global image information as effectively as deep learning approaches, which utilize global information to enhance feature utilization and continuously learn and incorporate additional feature information. Consequently, fragmented patterns are prevalent. Comparing the visual results of our method's extraction of aquaculture seas with those of classical machine learning, we can draw the conclusion that our approach is more global and comprehensive. Our method excels in effectively

balancing the supplementation of detailed information and the extraction of the overall geographical features.

Overall, the model proposed in this study demonstrates accurate recognition of aquaculture sea, enabling effective discrimination between aquaculture sea and other land features. Moreover, it exhibits higher recognition integrity for large coastal areas.

Furthermore, this study conducted a quantitative analysis of the model performance based on the aforementioned evaluation metrics. The results are presented in Table 2, with optimal values highlighted in bold. As shown, the proposed method outperformed the other state-of-the-art approaches in all metrics. The precision achieved was 96.17%, indicating a 1.41% improvement over the best alternative method. The recall reached 97.02%, exhibiting a 0.2% improvement. The overall accuracy reached 94.86%, demonstrating a 2.77% improvement. The mean intersection over union was 87.23%, revealing a substantial improvement of 6.67%. Finally, the $F1_{score}$ reached 96.56%, indicating a 1.66% improvement over the best alternative method. These results strongly support the superior performance of the proposed method in automatic recognition of aquaculture sea areas.

**Table 2.** Precision evaluation results of test areas (comparison experiments).

| Method | Precision (%) | Recall (%) | OA (%) | mIoU (%) | $F1_{score}$ (%) |
|---|---|---|---|---|---|
| Ours | **96.17** | **97.02** | **94.86** | **87.23** | **96.59** |
| SegNet | 93.11 | 96.53 | 92.03 | 80.56 | 94.79 |
| U-Net | 94.76 | 95.11 | 92.09 | 79.88 | 94.93 |
| DenseNet | 93.40 | 95.93 | 91.54 | 78.14 | 94.65 |
| SVM | 91.36 | 96.74 | 90.68 | 77.26 | 93.97 |
| RF | 89.99 | 96.82 | 89.52 | 74.50 | 93.28 |

## 6. Discussion

The importance of the sea for aquaculture has been given a progressively increasing amount of attention, so people have begun to pay more attention to its management and detection. Deep learning is one of the most popular methods for extracting aquaculture sea from remote sensing images, but the effectiveness of the method is closely related to the structure of the network. We added two modules to our network, PyConv and CBAM, and demonstrated the effectiveness of these two modules through an ablation study.

### 6.1. Visual Comparison

To validate the effectiveness of different modules in the proposed improved network, a series of experiments was conducted on the aquaculture sea dataset. These experiments involved the use of different modules, including the original SegNet network, SegNet network with the PyConv module, SegNet network with the CBAM module, and the improved SegNet network with both the PyConv and CBAM modules. The performances of these four models on the three test areas were evaluated through a comparative analysis, as shown in Figure 7.

Based on the visual comparison, the combination of the CBAM and PyConv modules yielded the best visual results. This combination improved the overall recognition of aquaculture sea areas in comparison with the original SegNet model or the models with a single module, as indicated by the green boxes in the figure. Furthermore, it significantly enhanced the recognition of finer drainage channels, as depicted by the red boxes. It also reduced misclassification, as illustrated by the yellow boxes.

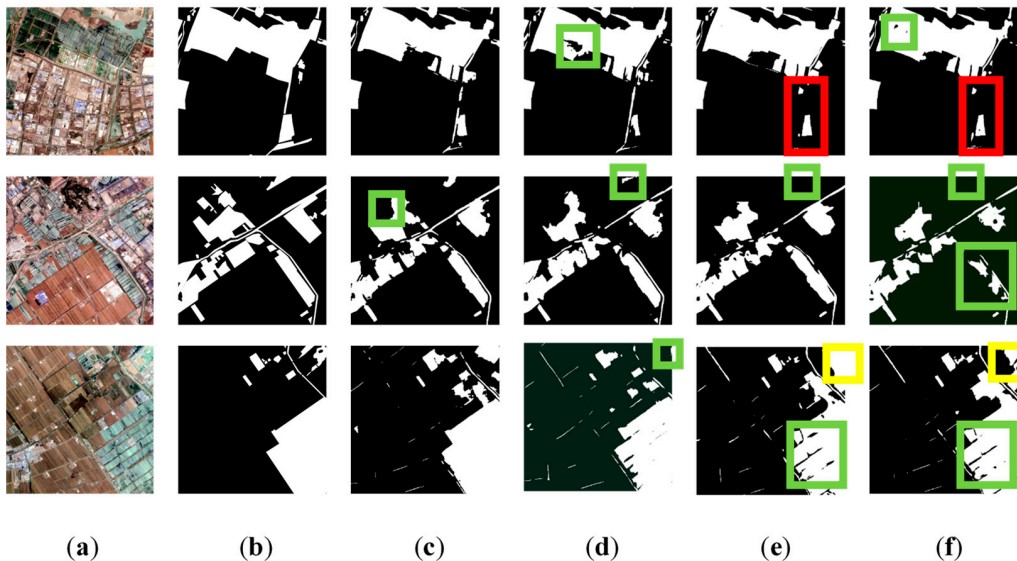

(**a**)   (**b**)   (**c**)   (**d**)   (**e**)   (**f**)

**Figure 7.** Comparison of identification results in the test area: test area 1 (row 1), test area 2 (row 2), and test area 3 (row 3). (**a**) Test images; (**b**) True labels; (**c**) Ours (SegNet + PyConv + CBAM); (**d**) SegNet + PyConv; (**e**) SegNet + CBAM; (**f**) SegNet.

*6.2. Precision Quantitative Analysis*

Furthermore, the quantitative analysis of the accuracy evaluation metrics, as presented in Table 3 with optimal values highlighted in bold, provided further insights. The data strongly demonstrated that both the CBAM and PyConv modules, when used individually, improved accuracy in comparison to the original model. However, their combination achieved the best performance, significantly enhancing the accuracy of automatic recognition of aquaculture sea areas. The improved model exhibited an improvement in overall accuracy of 2.83% compared to the original model, 2% compared to the model with the CBAM module alone, and 0.46% compared to the model with the PyConv module alone. Moreover, the mean intersection over union improved by 6.67%, 6.48%, and 1.33%, and the F1$_{\text{score}}$ increased by 1.8%, 1.07%, and 0.26%, respectively.

**Table 3.** Precision evaluation results of test areas (ablation study).

| Method | Precision (%) | Recall (%) | OA (%) | mIoU (%) | F1$_{\text{score}}$ (%) |
|---|---|---|---|---|---|
| SegNet + PyConv + CBAM | **96.17** | 97.02 | **94.86** | **87.23** | **96.59** |
| SegNet + PyConv | 94.87 | **97.83** | 94.40 | 85.90 | 96.33 |
| SegNet + CBAM | 93.46 | 97.68 | 92.86 | 80.75 | 95.52 |
| SegNet | 93.11 | 96.53 | 92.03 | 80.56 | 94.79 |

As demonstrated, the PyConv and CBAM modules produce model improvement. PyConv introduces an ability to address different sizes, which is like having multiple telescopes to observe images from different distances, so the model can obtain information at different scales, including details and overall features. This enables the model to understand the content of the image more fully and capture the characteristics of the image more comprehensively, thus improving the performance of the model in the task of aquaculture sea extraction. CBAM can be considered an "attention enhancer," which makes the computer vision model give more attention to the important objects in the image. It helps the model automatically find the channels in the image that are most important and suppresses the unimportant channels, thus facilitating a better understanding of the content of the image. At the same time, it also automatically finds the most important spatial positions in the image, gives greater attention to those important positions, and ignores the unimportant places, so the model can more accurately locate and identify the

targets in the image. Through the combination of these two functions, the CBAM module causes the model to adjust its attention at different scales automatically and prioritize the processing of important information, which enhances the performance of the model in the extraction of aquaculture sea.

In the future, we aspire to apply this technology to the management and monitoring of aquaculture seas, providing robust support to relevant agencies for timely and accurate acquisition of information regarding the area, distribution, and shape of aquaculture seas. With temporal data as supplementary information, our approach also holds the promise of offering a solution for detecting changes in aquaculture seas.

During the experimental analysis, we also found that the prediction results of the proposed model had a small number of missing points and misclassification problems. Further research will be performed to address these limitations. First, the segmentation accuracy for study areas with highly complex features needs to be improved further. Increasing the network depth of the proposed model and continuously improving the convolution module contact context information will help improve the accuracy for areas with particularly complex features. Second, introducing the spectral features of Landsat and other multispectral satellites into the depth learning method, which will combine more spectral information to obtain more accurate identification results of aquaculture seas, should be considered.

## 7. Conclusions

To achieve fast and accurate large-area automatic extraction of aquaculture seas, this study proposed a deep learning method based on an improved SegNet model. A pyramid convolution module was added to obtain multiscale information without additional network parameters, and the CBAM attention mechanism module was added to strengthen the use of effective information. The ablation experiment verified the effectiveness of the module. The combination use of the improved method was most helpful in improving accuracy. The overall accuracy, average cross-merge ratio, and $F1_{score}$ of the proposed model in identifying aquaculture seas in the three test areas improved by 2.83%, 6.67%, and 1.8%, respectively, compared with the original model. The effectiveness of the proposed model was verified by comparing it with the U-Net, SegNet, and DenseNet models, as well as traditional machine learning SVM and RF methods. The overall accuracy, average intersection ratio, and $F1_{score}$ of the proposed model in the three test areas were 94.86%, 87.23%, and 96.59%, respectively, which were better than those of the control method. The experimental results showed that the improved SegNet model proposed in this paper can automatically and accurately identify aquaculture sea areas and provide technical support for the monitoring and management of aquaculture seas. These findings highlight the practical relevance and importance of our enhanced SegNet model for automating precise aquaculture sea area identification. Our research offers promise for improving aquaculture sea monitoring and management, leading to more efficient decision-making.

**Author Contributions:** Conceptualization, methodology, writing—original draft preparation, W.X.; formal analysis, investigation, Y.D.; software, validation, writing—review and editing, X.R.; resources, data curation, visualization, supervision, project administration, Y.Z. (Yarong Zou) and Y.Z. (Yating Zhan). All authors have read and agreed to the published version of the manuscript.

**Funding:** This research was funded by the Jiangsu Province Marine Science and Technology Innovation Project, grant number JSZRHYKJ202207.

**Data Availability Statement:** Not applicable.

**Conflicts of Interest:** The authors declare no conflict of interest.

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
