# Peer review of "Automatic Extraction Method of Aquaculture Sea Based on Improved SegNet Model"

_water, doi:10.3390/w15203610_

Round 1

Reviewer 1 Report

The authors have introduced a novel approach that has clear implications for real-world problems and the practical relevance of the proposed approach is evident and opens up new avenues for solving problems of aquaculture seas’ automatic extraction. The manuscript is generally good, but there is still some room for improvement. Some of the grammar and language of the manuscript should be thoroughly enhanced. It is recommended to accept with minor revisions. Please find my additional recommendations below.

Comment 1. The experiment set by the author has been relatively complete, can you compare the method in this paper with the classic machine learning experiment, and then make a more detailed analysis?

Comment 2. Please revise the conclusion in paragraphs. Conclusions are not just about summarizing the key results of the study, it should highlight the insights and the applicability of your findings for further work. In the conclusions, in addition to summarizing the actions taken and results, please strengthen the explanation of their significance.

Comment 3. The discussion section of the article should include some discussion of future work, such as the future application of this work.

Comment 4. In the fourth paragraph of Introduction section, this sentence “The object-oriented classification method has some limitations. Its accuracy de-pends heavily on the segmentation scale and parameters, but it is often difficult to determine the optimal value and it needs to be adjusted repeatedly.” describes the limitations of object-oriented classification methods, please put it at the end of the third paragraph, and it is more closely related to the above.

Comment 5. Please improve the resolution of Figs. 1 and 3.

Comment 6. In the Introduction section, the last paragraph does not describe what problem you are focused on solving, but simply describes your approach. Please supplement and improve it.

“Therefore, based on remote sensing imagery of the GF-1D satellite and to improve the automatic extraction accuracy of aquaculture sea and provide technical support for scientific monitoring and management of aquaculture sea, this study constructed an improved SegNet model by introducing the pyramid convolution module and attention mechanism module.”

Comment 7. In the 2.1. Study Area section, please highlight why you chose this study area and what characteristics of the study area are important factors.

Comment 8. In the 2.3. Data Preprocessing section, the sentence "Consequently, 10,000 sample pairs, each consisting of a 256 × 256 pixel image and its corresponding label, were generated. "should be amended to" Consequently, 10,000 sample pairs, each consisting of a 256 × 256 pixels image and its corresponding label, were generated. ", because pixels should be used in complex units here, please correct.

Comment 9. Please add a separator for the numbers over 1,000. Check all numbers including those in the tables. Check all format, e.g. F1 score in all the manuscript, score should be in subscript like F1score, check all.

Comment 10. In the 3.1. Experimental Platform and Parameter Settings section, the manuscript does not mention the adjustment and selection of parameters. Please explain why you chose 200 for the epoch. Is it possible to achieve good training results with a smaller epoch? In addition, how do you determine the size of the batch size and learning rate parameters? Please add clarification.

Author Response

Thanks for your suggestion. Please check the attached file.

Reviewer 2 Report

First of all, I want to congratulate the authors for their efforts in this manuscript. They present the use of unsupervised extraction methods to identify aquaculture sea-based using an improved SegNET Model. The topic is very interesting and aligned with the journal's scope. In general terms, the results of their proposal are adequate, and the paper is well-written. Nonetheless, there are some issues to be solved, the most relevant one linked to the structure of the paper. Thus, a series of comments are included below aimed at enhancing the quality of the paper:

1.       Consider avoiding the use of acronyms in the abstract, especially for those acronyms used once or twice. If the acronym is necessary, define it the first time it is used.

2.       The term aquaculture sea, defined at the beginning of the introduction, needs to be referenced since it is not easy to find this term in the bibliography, as the authors have presented in the introduction.

3.       Part of the introduction must be moved to a new section named Related Work (just after the introduction). In this new section, the authors have to describe current solutions for the extraction of aquacultural areas using remote sensing. So far, in the introduction, the authors included the references [6] to [27], which can be sufficient for the related work.

4.       The remaining introduction needs to be improved. After the first paragraphs, the authors must detail the benefits of using extraction based on remote sensing for aquaculture and mention the limitations of this technique and the main complements for this (such as in-situ sensors). Then, the authors have to provide a summary(about 200 words) of the current solutions in the introduction as the third paragraph of the introduction.

5.       The aim of the paper must be presented in an independent paragraph at the end of the introduction. The authors have to highlight the most important novelties and contributions of their proposal. The current introduction presents the aim of the paper in lines 108-112. It must be extended considering the issues mentioned above.

6.       Figure 1 a and b represents the same area. It would be better if Fig 1 a represents the location of the province in the country. Please add the north arrow to the different subfigures. In subfigure c, I suggest changing remote sensing image to truecolour image.

7.       In subsection 2.2, It is suggested to authors to include more information about the used satellite, such as spectral, temporal, and radiometric resolution.

8.       The content of 3.1. Experimental Platform and Parameter Settings and Precision Evaluation Metrics cannot be in the results section since they describe the methodology. I suggest creating a new section named Methodology and including this content in the new section.

9.       In Figure 7, it is recommended to include a scale and the north arrow below the images (if all of them have the same scale or at the left of the images if they are different.

10.   Subsection 1.1. Prior Work must be an entire section after the results.

11.   The discussion section should be divided into different subsections presenting each key aspect of their research. Moreover, the authors must compare their results with existing results in the literature, including a comparative table if possible. Other aspects to be analyzed include the impact of these results on aquaculture management and the limitations of conducted research. 

Round 2

Reviewer 2 Report

The comments were correctly addressed by the authors and the paper is now ready to be accepted.